# Unraveling the Cave: A Seventy-Year Journey into the Caveolar Network, Cellular Signaling, and Human Disease

**DOI:** 10.3390/cells12232680

**Published:** 2023-11-22

**Authors:** Alessio D’Alessio

**Affiliations:** 1Sezione di Istologia ed Embriologia, Dipartimento di Scienze della Vita e Sanità Pubblica, Università Cattolica del Sacro Cuore, 00168 Roma, Italy; alessio.dalessio@unicatt.it; 2Fondazione Policlinico Universitario “Agostino Gemelli”, IRCCS, 00168 Rome, Italy

**Keywords:** calcium signaling, caveolae, caveolin-1, cell senescence, ECs, eNOS, endocytosis, lipid rafts, signal transduction

## Abstract

In the mid-1950s, a groundbreaking discovery revealed the fascinating presence of caveolae, referred to as flask-shaped invaginations of the plasma membrane, sparking renewed excitement in the field of cell biology. Caveolae are small, flask-shaped invaginations in the cell membrane that play crucial roles in diverse cellular processes, including endocytosis, lipid homeostasis, and signal transduction. The structural stability and functionality of these specialized membrane microdomains are attributed to the coordinated activity of scaffolding proteins, including caveolins and cavins. While caveolae and caveolins have been long appreciated for their integral roles in cellular physiology, the accumulating scientific evidence throughout the years reaffirms their association with a broad spectrum of human disorders. This review article aims to offer a thorough account of the historical advancements in caveolae research, spanning from their initial discovery to the recognition of caveolin family proteins and their intricate contributions to cellular functions. Furthermore, it will examine the consequences of a dysfunctional caveolar network in the development of human diseases.

## 1. Introduction

Caveolae are small, flask-shaped invaginations in the cell membrane that belong to a specialized subgroup of non-planar (invaginated) lipid rafts in the cell plasma membrane of eukaryotic cells, involved in various cellular processes. Lipid rafts and caveolae are both specialized cell membrane microdomains, but they have distinct characteristics and functions. Both structures are enriched in specific lipid components, particularly cholesterol and sphingolipids, that make these membrane regions more ordered compared to the surrounding lipid bilayer. Lipid rafts are flat, smaller, less organized, and more fluid in nature compared to caveolae [1]. They serve as central hubs for bringing together signaling molecules, which are essential for facilitating favorable interactions required for signal transduction. Conversely, caveolae have a well-defined, stable, and rigid structure, owing to the presence of distinct scaffolding proteins that grant them their characteristic omega-shaped appearance [2]. According to electron microscopy studies, caveolae have a diameter ranging from 60 to 80 nm, with a narrow neck of 10 to 50 nm. These microscopic structures serve crucial roles in numerous physiological functions, including the regulation of lipid metabolism, intricate intracellular signaling adjustments, and participation in vital processes such as membrane repair, mechanosensation, and cellular trafficking [3,4,5,6]. Furthermore, their involvement in a variety of pathologies, such as cancer, cardiovascular disorders, and neurodegenerative conditions, offers intriguing prospects for future research [7,8,9]. This article seeks to provide a concise overview of pivotal discoveries in the caveolae field, highlighting their profound significance in cellular biology and their potential as promising therapeutic targets for a spectrum of human diseases.

## 2. Historical Milestones in Caveolae Research

The study of caveolae has seen several significant milestones since their discovery, enhancing our comprehension of these unique membrane features (Figure 1). In the mid-1950s, early electron microscopy investigations into cell membranes in continuous endothelium by George Emil Palade unveiled characteristic striated flask-shaped structures, which he referred to as “plasmalemmal vesicles” [10]. In 1953, Eichi Yamada found similar features in the gall bladder epithelium and named them “caveolae intracellulares” due to their resemblance to “little caves” [11]. From the time they were first identified, increasing numbers of studies have consistently detected caveolae-like structures in many cell types, including adipocytes, endothelial cells (ECs), fibroblasts, muscle cells, and epithelial cells [12,13,14,15]. ECs show a high prevalence of caveolae, and in adipocytes, they can make up 30 to 50% of the cell surface [16]. However, some eukaryotic cells, including red blood cells (erythrocytes), lymphocytes [17], and some epithelial cells [18], do not possess these structures. In the nervous system, the presence of caveolin proteins had remained elusive for a long time, contributing to the prevailing notion that caveolin proteins were absent in brain tissue and neurons. The perspective shifted in the late 1990s with the discovery of caveolins in different types of brain cells, including brain ECs, glial cells, and neurons [19,20,21]. It is noteworthy that neurons seemingly express all three caveolins, although certain authors concur that they might be devoid of caveolae, thus implying a potential function for these proteins beyond the confines of the plasma membrane caveolae [22,23]. These hypotheses find support in studies, indicating that caveolins can be found in both buoyant and heavy fractions resulting from density gradient centrifugation, implying a more extensive distribution than solely within caveolae. While not the focus of this review, it is important to note that non-caveolar caveolins act as versatile regulators of cellular functions by facilitating interactions with proteins excluded from typical caveolae and influencing various cellular processes [24]. For instance, these caveolins are believed to play a crucial role in determining intracellular lipid fluxes, including cholesterol and sphingolipids, thereby directly or indirectly modulating lipid-dependent processes. Insights from invertebrate systems underscore that the loss of function in these caveolins is associated with common phenotypes and pathologies observed in caveolin-deficient cells and animals [25]. The consideration of these findings is imperative when conducting investigations focused on endogenous caveolin protein expression, such as through RNA interference (RNAi). In such scenarios, it becomes challenging to distinguish between the effects of caveolins localized within caveolae and those found outside, i.e., in cells that exhibit caveolar structures. In the ever-evolving landscape of cellular biology, the continuous investigation of the caveolar network promises to reveal new insights into the complex mechanisms that govern cell function and has significant implications for our understanding of human diseases and their treatment.

## 3. Caveolins and Cavins: Synthesis and Post-Translational Modifications

### 3.1. Caveolin-1

The first recognized protein marker of caveolae, called caveolin-1, was not discovered until the early 1990s (Figure 1). This finding offered valuable insights into the structural and functional characteristics of caveolae [13]. The same year Glenney cloned and sequenced a human cDNA encoding the caveolin-1 (CAV1) gene [26], translating a full-length protein of 178 amino acids that was expressed on the surface and in the cytoplasm of ECs of pulmonary arteries [27]. This was a pivotal discovery, as this monotopic membrane protein was found to be a major structural coat protein playing a crucial role in caveolae biogenesis [28]. Each caveolin monomer consists of distinct regions, which include: (i) a cytosolic N-terminal domain that faces the cytoplasm; (ii) a C-terminal domain, encompassing residues 135 to 178; (iii) a transmembrane domain (TMD) spanning residues 102 to 134; and (iv) an oligomerization domain (OD), spanning residues 61 to 101, that incorporates both the highly conserved “FEDVIAEP” caveolin signature motif (CSM, residues 68 to 75) and the caveolin scaffolding domain (CSD, residues 82 to 101) [29,30,31,32] (Figure 2). Notably, a cholesterol recognition/interaction amino acid consensus (CRAC) “VTKYWFYR”, which spans residues 94–101 within the CSD [33], allows caveolin-1 to interact with cholesterol molecules. A more recent study, employing cryo-electron microscopy, has significantly enhanced our understanding of the tridimensional organization of caveolin-1. The study unveils that the human caveolin-1 complex comprises 11 protomers, which are intricately arranged to construct a densely packed disc. This disc encompasses an outer portion, a central β-barrel domain, and 11 curved α-helical domains. These structural insights offer a fresh perspective on the involvement of crucial regions in human caveolin-1, including scaffolding, oligomerization, and intramembrane domains, in the process of membrane remodeling [34]. Two isoforms of caveolin-1 have been elucidated, referred to as caveolin-1α (including residues 1–178) and caveolin-1β (spanning residues 32–178). While the precise role of these two protein variants in caveolae formation is still largely obscure, it appears that the α isoform demonstrates greater efficacy in driving the genesis of caveolae compared to the β isoform [35]. Caveolin-1 undergoes post-translational modification (PTM) through palmitoylation, which involves the covalent attachment of palmitic acid to three distinct cysteine residues within the TMD domain (Figure 2). The palmitoylation of proteins by palmitoyl transferases is generally a reversible process, playing a crucial role in regulating various cellular functions. This includes the control of protein localization, membrane organization, and signal transduction processes, ultimately contributing to the overall stability and functionality of the cell. Studies involving a palmitate-deficient mutant of caveolin-1 have revealed that the palmitoylation of caveolin-1 is not essential for its localization to caveolae [36] and is irreversible in ECs [37]. Caveolin-1 can also undergo phosphorylation [38], occurring both at tyrosine 14 and at serine 80 (Figure 2). Although the functional implications of caveolin-1 phosphorylation have been relatively understudied, the two phosphorylation sites seem to serve distinct purposes. The tyrosine phosphorylation of caveolin-1 can be facilitated by various kinases, including those in the Src family, and it can influence the response to oxidative stress, promote cell senescence in chondrocytes, and modulate autophagy [39,40,41]. In contrast, the phosphorylation of the serine residue in caveolin-1 influences its cellular localization within the endoplasmic reticulum, directing it towards the secretory pathway [42]. Ubiquitination can also regulate the expression of caveolin-1. When caveolae assembly is dysfunctional or the caveolin-1 levels increase, it can be ubiquitinated and subsequently degraded within lysosomes [43]. More recently, it has been shown that the degradation of caveolin-1 by the E3 ubiquitin ligase ZNRF1 is vital for the regulation of the TLR4-mediated inflammatory pathway [44].

### 3.2. Caveolin-2

In 1996, caveolin-2, the second member of the caveolin family, was identified and cloned by Scherer and collaborators [30]. Caveolin-2 forms hetero-oligomers with caveolin-1 [2], and both proteins are found typically co-expressed in the same cells [30,45]. Unlike caveolin-1, the lack of caveolin-2 does not affect the formation of caveolae. However, in specific cell types, the concurrent presence of both proteins appears to enhance caveolae formation more effectively than caveolin-1 alone [35]. Like caveolin-1, caveolin-2 shows multiple isoforms characterized by distinct but partially overlapping subcellular distributions [46]. These include the full-length caveolin-2α and two truncated variants known as caveolin-2β and caveolin-2γ. Nonetheless, the functional significance of these caveolin-2 isoforms remains largely unknown [47]. Like caveolin-1, caveolin-2 undergoes post-translational modifications, including myristoylation at a single glycine residue and palmitoylation at cysteine 109, 122, and 145, which mainly control protein localization in the plasma membrane (Figure 3). Recently, it has been revealed that the acylation of caveolin-2 directs the protein to be phosphorylated by the insulin receptor tyrosine kinase, emphasizing its potential role in regulating insulin signaling [48]. Caveolin-2 can be also phosphorylated at serine 23 and 36, and this modification is crucial to sustain caveolin-1 activity during caveolae biogenesis [49]. Two additional sites of phosphorylation, namely tyrosine 19 and 27, have been identified within caveolin-2 that play crucial function in regulating cell signaling by promoting protein interaction with SH2 bearing domains [50,51] and are involved in adipocyte hypertrophy [52].

### 3.3. Caveolin-3

The third member of the caveolin gene family, termed caveolin-3, was discovered in 1996 by Tang and colleagues [31]. Caveolin-3 is predominantly expressed in mature smooth, cardiac, and skeletal muscle cells [53]. Like caveolin-1, caveolin-3 shows distinct structural domains, including (i) an N-terminal domain, spanning residues 1–54, which includes the FEDVIAEP CSM (residues 41–48); (ii) a CSD spanning residues 55–74; (iii) a TMD (residues 75–106); and (iv) a C-terminal domain (residues 107–151, facing the cytoplasm) (Figure 3) [54]. Caveolin-3 is also subjected to post translation palmitoylation at three different cysteine residues [55]. While caveolin-3 is shorter in length compared to caveolin-1, it exhibits significant structural and functional similarities to caveolin-1 and possesses the ability to independently form caveolae [30]. The caveolin-3 (CAV3) gene encodes an open reading frame of 151 amino acids that is 65% identical and 85% similar to caveolin-1. In addition to palmitoylation, caveolin-3 undergoes post-translational modification through SUMOylation, which is a process involving the attachment of small ubiquitin-like modifier (SUMO) to its N-terminus. This modification of caveolin-3 plays a crucial role in regulating the expression of β-adrenergic receptors [56]. Mutations in CAV3 gene lead to a series of a family genetic disorders known as caveolinopathies [57], leading to rare forms of muscle dysfunction, such as muscular dystrophies, myopathies, and arrhythmias. Four different caveolinopathies have been identified, including limb–girdle muscular dystrophy (LGMD) 1C, rippling muscle disease (RMD), isolated hyperCKemia, and distal myopathy [57,58]. Within skeletal myofibers caveolin-3 takes part in the dystrophin-glycoprotein complex (DGC), serving as a link between the extracellular matrix and intracellular cytoskeletal elements. The connection between caveolin-3 and DGC within muscle cells plays a pivotal role in upholding the structural integrity of the sarcolemma, enabling the normal functioning of muscle cells [53]. Although the post-translational modifications of caveolin-3 have received limited attention, there is evidence to suggest that mutations in the CAV3 gene, which result in muscular dystrophy, are broken down through ubiquitination and proteasomal pathways. Blocking proteasomal degradation could potentially restore normal caveolin-3, which holds promise for the treatment of LGMD-1C from a clinical perspective [58].

### 3.4. The Cavin Family of Proteins

In the 2000s, the identification of novel structural constituents of caveolae, known as caveolae-associated proteins (cavins), has augmented our comprehension of the intricate architecture of caveolae [59,60,61] (Figure 1). Cavin proteins, including cavins 1–4, encoded by the polymerase I and transcript release factor (PTRF), serum-deprivation-response protein (SDPR), SDPR-related gene product that binds to C-kinase (SRBC), and muscle-related coiled-coil protein (MURC) genes, respectively, assume a pivotal role in the biogenesis of caveolae. Among cavin proteins, only cavin-1 appears crucial for the correct biogenesis of caveolae [62], while the other three members of the family have been suggested to contribute to caveolae stabilization [63,64,65] and appear to display tissue-specific distributions. Their intricate interplay with caveolin proteins and their involvement in membrane dynamics impart this family of proteins with essential significance in many cellular processes and signal transduction mechanisms associated with the presence and functionality of caveolae [66,67]. The structure and function of the cavin family of proteins have been reviewed by Kovtun et al. [67]. Notably, aside from its involvement in caveolae formation, it has come to light that patients with cavin-1 mutations manifest disorders, including lipodystrophy and muscular dystrophy, affecting organs such as adipose tissue, skeletal muscle, heart, and lungs. Nevertheless, the molecular mechanisms underlying the involvement of cavin-1 in these disorders remain largely unclear and necessitate thorough investigation [68].

## 4. Functional Roles of Caveolae and Caveolins

### 4.1. Caveolae and Caveolin in Endocytosis and Transcytosis

Endocytosis is a fundamental cellular process through which cells internalize molecules, particles, or fluids from their external surroundings for various purposes, including nutrient uptake, receptor internalization, and the removal of waste. This cellular uptake mechanism encompasses three primary forms: phagocytosis (the engulfment of large particles), pinocytosis (the taking up of small, dissolved molecules and fluids), and receptor-mediated endocytosis [69]. In turn, transcytosis refers to the transport of materials from one side of the cell to the other (e.g., from the apical to the basolateral membrane or vice versa), thereby making them accessible to neighboring cells and different tissue compartments. It is particularly involved in polarized epithelial cells that line body surfaces like endothelia and intestinal epithelium. Functionally, transcytosis combines endocytosis (uptake at one membrane) and exocytosis (release at the opposite membrane) [70,71].

The strategic positioning of caveolae in the plasma membrane, their unique shape and composition, and the ability of their scaffolding proteins to interact with multiple molecules all make them ideal for aiding in the uptake of substances into the cell. Therefore, in addition to the well-known endocytic pathway mediated by clathrin-coated vesicles (CCV) [72], caveolae-dependent endocytosis may stand for an alternative high specific component of cellular uptake. Yet, to date, the contribution of plasma membrane caveolae to endocytosis is still debated. In initial investigations pertaining to the participation of caveolae in endocytosis, experiments utilizing a caveolin-1 construct tagged with green fluorescent protein (GFP) indicated that caveolae anchored to the plasma membrane exhibit relative immobility [73]. Furthermore, caveolin-1 has been postulated to function in the stabilization of caveolae-like invaginations on the cell surface, thereby acting as a rate-limiting factor in the internalization process of these organelles [74]. In this context, the localization of the dynamin-related EH domain-containing protein 2 (EHD2), situated in the caveolar neck, may serve to enhance the stabilization of caveolae at the plasma membrane. Indeed, its overexpression has been observed to result in an increased number of static surface caveolae [75]. Like EDH, Pacsin2, a protein that regulates the morphogenesis and endocytosis of caveolae, plays a crucial role in maintaining the proper functioning and stability of caveolae on the cell membrane. Its loss or deletion may lead to the detachment of caveolae from the cell surface [76]. On the contrary, akin to lipid rafts, caveolae exhibit the capacity to translocate within the confines of the plasma membrane while preserving the integrity of their caveolar coat [77]. It is imperative to clarify that, although caveolae can detach from the plasma membrane, leading to their internalization and trafficking to intracellular organelles, the underlying mechanism and the involved cargoes remain unclear. This lack of clarity has given rise to controversy within the field. Therefore, while clathrin-mediated endocytosis is a thoroughly characterized mechanism known for its precise cargos and intracellular pathways, the involvement of caveolae in endocytosis remains a challenging issue [75,78]. It has been suggested that detached caveolae have the capacity to undergo fusion with the early endosome. Subsequently, the maturation process ensues, progressing from early endosomes to late endosomes and multivesicular bodies, culminating in the ultimate degradation of the caveolae contents within lysosomes. In this context, empirical evidence indicates that caveolin-1 exhibits co-localization with both early and late endosomal markers, notably Rab5 and Rab7 [43,78]. Furthermore, it has been observed that the immunogold labeling of caveolin-1 accumulates in multivesicular bodies after detachment from the plasma membrane [79]. Studies have proposed that the internalization of caveolae depends on the phosphorylation of caveolin-1 and caveolin-2. This assertion is supported by experiments conducted in the presence of phosphatase inhibitors, which markedly augment this process [80,81]. Caveolin-1-deficient mice cannot internalize albumin compared to their wild-type counterparts, supplying further confirmation of the integral role played by this protein in transcytosis [82]. Molecules that have suggested to undergo internalization via caveolae include albumin, folic acid, alkaline phosphatase, lipids, insulin, low-density lipoproteins (LDL), chemokines [83,84,85,86,87], and certain pathogens such as toxins, viruses and bacteria [88,89,90,91,92,93,94,95,96,97,98,99]. In conclusion, the pivotal roles of the caveolar network in cellular uptake suggest that targeting these components could hold great promise for advancing therapies in drug delivery and disease treatment. In this context, it is of paramount importance to identify molecules resident in caveolae that are exclusively internalized through caveolae-mediated endocytosis.

### 4.2. Caveolins in Cell Signaling

The “caveolae signaling hypothesis” proposed in the mid-1990s (Figure 1) is a concept related to the discovery of the caveolin family of proteins and their role in cellular signaling [100]. The hypothesis relies upon the presumption that signaling proteins, possessing a distinctive peptide sequence referred to as the caveolin binding motif (CBM), can establish interactions with the CSD. The hydrophobic consensus, rich in aromatic residues, for CBM is βXXXXββ and βXβXXXXβ, where β represents aromatic amino acids (Tryptophan, Phenylalanine, and Tyrosine) and X denotes any amino acid. It is widely acknowledged that, following CSD–CBM interaction, caveolin can maintain the signaling molecules in an inactive status, until the interaction is disrupted by specific cues. Remarkably, although this mechanism has garnered broad acknowledgment as a primary regulatory process for signaling proteins by caveolins, contemporary structural and bioinformatic analyses calls it into question [101,102]. Consequently, certain authors advocate for a re-evaluation of the functional significance of interactions dependent on CBM/CSD in regulating the functions of signaling molecules. It has been suggested that only a small amount, namely about 30%, of signaling molecules would express a CBM, regardless of its cellular distribution [102]. Moreover, critiques have been raised concerning experiments that entail the association of signaling proteins with caveolins through immunoprecipitation, primarily ascribed to the restricted solubility of caveolin-enriched domains [2]. Notably, numerous studies have suggested that interactions between caveolin and target proteins do not invariably require the participation of CSD and CBM [103]. In certain instances, protein interactions may also encompass multiple domains of caveolin-1, including the C-terminal domain [104,105]. Nevertheless, whether a direct or indirect interaction exists between CSD and CBM, there is no doubt that it facilitates the spatial regulation of various signaling pathways. Consequently, it can be anticipated that any dysregulation of this mechanism, as well as the deletion or mutation of CSD and CBM, may have implications for various physiopathological processes or disease states.

Caveolin-1 has been shown to interact with RTKs like the epidermal growth factor receptor (EGFR) [106] and insulin receptor (IR) [107,108], thereby modulating their downstream signaling. These interactions have significant implications for cellular proliferation and survival. Caveolins also influence G-protein-coupled receptors (GPCRs)-mediated signaling [109]. The role of caveolin-1 in GPCR endocytosis and desensitization is well-established, impacting processes ranging from neurotransmission to immune responses. Additional signaling molecules can be regulated through their confinement within the caveolae network or interaction with caveolin-1 [110,111,112,113,114,115,116,117]. In the context of endothelial function, caveolins play a crucial role in regulating nitric oxide (NO) signaling. Caveolin-1 physically interacts with endothelial NO synthase (eNOS), restricting its enzymatic activity and preventing the untimely production of NO. In situations of physiological stimulation, such as increased shear stress or specific agonists, caveolin-1 can dissociate from eNOS, allowing the enzyme to become activated and generate NO. This dynamic interaction between caveolin-1 and eNOS serves as a crucial mechanism for maintaining vascular homeostasis, as NO serves as a potent vasodilator and regulator of blood pressure, impacting various physiological and pathological processes, including cardiovascular health [118,119,120]. In addition, the post-translation phosphorylation of caveolin-1 can affect downstream signaling. Furthermore, it is noteworthy that the post-translation phosphorylation of caveolin-1 has the potential to influence downstream signaling. For instance, the phosphorylation of caveolin-1 plays a crucial role in regulating autophagy under the conditions of oxidative stress and cerebral ischemic injury, resulting in protecting against cellular damage caused by oxidative stress, and potentially reducing cerebral infarct damage during ischemic events [40]. Moreover, the tyrosine phosphorylation of caveolin-1 in ECs exposed to oxidative stress or alterations in flow conditions can govern both cell permeability and mechanotransduction responses [121,122], and its inhibition hinders Rac1/Cdc42-mediated axonal growth in human neuronal progenitor cells [123]. Like caveolin-1, caveolin-2 phosphorylation plays a critical role in the regulation of signal transduction, specifically in the context of cellular communication and intracellular signaling. This post-translational modification of caveolin-2 has been shown to impact its interactions with various signaling molecules [124]. In summary, it is imperative to conduct a meticulous assessment of the nature and significance of the “caveolae signaling hypothesis” by leveraging state-of-the-art bioinformatic tools. The incorporation of these methodologies in conjunction with biochemical investigations is certain to yield significant contributions to comprehending the role of caveolins in the regulation of signal molecules.

### 4.3. Caveolins in Host Cell Response and Inflammation

Inflammation is the body’s response to infections and tissue damage in vascularized tissues. The function of this process is to transfer cells and molecules responsible for host defense from the circulation to the injured sites, ultimately leading to the elimination of the offending agents. It is a complex and tightly regulated physiological response triggered by the host immune system to combat various harmful stimuli. The typical inflammatory reaction involves a series of coordinated steps, including the recognition of harmful stimulus (e.g., injury, infection, or tissue damage); the recruitment of white blood cells, particularly neutrophils and macrophages, to the site of inflammation; leukocyte activation; and the resolution of inflammation. Interestingly, the caveolar network can take part in these events. First, pathogens like viruses and bacteria can use all the molecules found in caveolae, which are essential for vesicle formation and fusion, to enter the host cell [81,94,95,125,126,127,128,129,130,131,132,133,134]. Of note, since caveolae do not follow the typical path to lysosomes, pathogens that enter through these structures can remain viable and avoid being broken down in lysosomes [135]. A crucial aspect of inflammation involves leukocyte adhesion to ECs [136]. During this process, leukocytes first interact with adhesion molecules on EC surface, such as E- and P-selectin, and later with intercellular adhesion molecule-1 (ICAM-1) and vascular cell adhesion protein 1 (VCAM-1). Interestingly, there are reports indicating that ICAM-1 plays a role in helping white blood cells exit the bloodstream by moving to areas rich in caveolin-1 within the lateral portion of ECs. This relocation of ICAM-1 is believed to promote the formation of pseudopodia by leukocytes, facilitating their exit into the surrounding tissue [137,138]. A study by Michell et al. found that exposure to high pressure led to increased leukocyte adhesion and that inflammation in rat carotid arteries is triggered by the mechanosensory capability of caveolin-1. Consequently, mice and ECs lacking caveolin-1 were protected from pressure-induced inflammation, highlighting the significance of caveolin-1 in pressure-induced vascular inflammation [139]. Supporting the role of the caveolar network in recruiting white blood cells, it has been shown that caveolin-1 reduces excessive NO-related permeability in blood vessels and facilitates white blood cell adhesion through a specific mechanism involving ICAM-1 and Src activation [140]. Besides caveolin-1, caveolin-2 has been also found to facilitate leukocyte infiltration in brain ECs, contributing to the development of common neurodegenerative diseases [141]. In addition, this protein has been suggested to play a protective role in reducing postischemic tissue injury by dampening plasminogen activator inhibitor-1 (PAI-1) expression, leading to reduced leukocyte recruitment by ECs. Authors suggest that targeting PAI-1 or enhancing caveolin-2 expression could be potential therapeutic strategies to reduce tissue damage caused by ischemia/reperfusion injury [142]. These findings highlight the intricate involvement of caveolae and caveolins in the inflammatory response and underscore their vital role in regulating cellular signaling and immune processes.

### 4.4. Caveolin Proteins and Mechanotransduction

Mechanotransduction, the process by which cells sense and respond to mechanical forces, plays a pivotal role in numerous physiological and pathological processes [143,144,145]. Caveolins not only modulate intracellular signaling cascades [146] but influence the mechanical properties of the plasma membrane itself [7]. These features make caveolins crucial players in the ability of cells to sense and respond to mechanical cues, which is fundamental to various physiological processes, including tissue development, vascular homeostasis, and cancer metastasis [147]. Caveolin-1 is the most extensively studied isoform in this context, thanks to its capability to regulate and organize signaling molecules in caveolae, facilitating their rapid response to mechanical stimuli. Its involvement in mechanosensing has been evidenced in several studies, such as one by Drab et al., that highlighted the role of caveolin-1 in the activation of eNOS in ECs in response to shear stress [148]. Furthermore, caveolins can regulate the activation of integrins and focal adhesion kinase, influencing cell adhesion and migration in response to mechanical cues [149]. In ECs and muscle cells, caveolae can serve as a repository for membranes, which can undergo flattening when the cell experiences mechanical stress [149]. Therefore, unstimulated cells have excess membrane confined within invaginated caveolae, granting them the ability to alleviate plasma membrane tension, thereby effectively counteracting mechanical stress. In such a scenario, the reconfiguration of the plasma membrane after the breakdown of caveolae would thereby facilitate the release/activation of mechanoreceptors that had been previously trapped within the caveolae. More recently, Moreno-Vicente and colleagues have demonstrated that caveolin-1 can regulate the activity of the YAP (Yes-associated protein) and TAZ (transcriptional coactivator with PDZ-binding motif), two central components of the Hippo signaling pathway, which governs cell proliferation, organ, and mechanosensing [150,151]. The recent identification of caveolin-1 invaginations, distinct from conventional caveolae, and devoid of cavins, referred to as “dolines”, has introduced a new layer of complexity to our knowledge of mechanotransduction [152]. These structures denote distinct plasma membrane invaginations regulated by caveolin-1 in the absence of caveolae, playing a role in the perception of mechanical forces. The authors suggest that these structures are capable of sensing forces within the low to medium range, thereby imparting consistent mechanoadaptation and mechanoprotection to tissues lacking caveolae. Therefore, dolines, alongside caveolae, function distinctly and complementarily to mediate the cell responses to mechanical stimuli originating from the extracellular environment. In summary, elucidating the precise mechanisms by which caveolins contribute to mechanotransduction is a growing area of research, with implications for understanding cell biology, disease pathogenesis, and potential therapeutic interventions.

### 4.5. Caveolae in the Regulation of Calcium Signal

Calcium ion (Ca^2+^) is perhaps the most prevalent and adaptable intracellular messenger across various cell types [153]. Intracellular (Ca^2+^) signaling plays a pivotal role in a multitude of cellular processes, ranging from muscle contraction and neurotransmission to cell proliferation and gene expression. The precise control of Ca^2+^ concentration within various cellular compartments is essential for these diverse functions, and cells have developed an intricate network of regulatory mechanisms, among which caveolae have emerged as a critical component [2]. The first indications of the involvement of caveolae in mobilizing intracellular Ca^2+^ come from studies conducted by Popescu et al. in the mid-1970s [154]. The authors showed that following the treatment of smooth muscle cells with calcium oxalate, the compound was found to be distributed in various cellular compartments, including invaginated caveolae. Further studies have demonstrated the localization of essential signaling molecules regulating Ca^2+^ mobilization, such as the inositol-1,4,5-trisphosphate receptor (IP3R) and Ca^2+^-ATPase, within caveolae both in ECs and smooth muscle cells [155]. In 1998, Isshiki demonstrated that caveolin-rich cell edges are involved in the initiation of ATP-induced Ca^2+^ waves in bovine aortic ECs [156], and two years later, the same group utilized a modified calcium sensor to illustrate the central function of caveolae in facilitating thapsigargin-induced Ca^2+^ entry [157]. They reported that interfering with these domains using a cholesterol-extracting agent impeded this process. In ECs, caveolin-1 expression is necessary for the relaxation associated with endothelium-derived hyperpolarizing factor (EDHF), as it influences the membrane positioning and functioning of transient receptor potential vanilloid (TRPV) channels and connexins, two crucial players in the EDHF-signaling pathway [158]. More recently, Medvedev et al. demonstrated the role of caveolae in regulating Ca^2+^ signaling in atrial cardiomyocytes and that their disruption leads to an increase in cAMP levels, which in turn enhances the phosphorylation of Protein Kinase A (PKA)-mediated Ca^2+^ signaling [159]. The specific isoforms of protein kinase C (PKC) are activated by calcium, thereby regulating diverse cellular functions, including cell growth, differentiation, and apoptosis. Among the different isoforms of PKC, PKCα has been shown to be mainly localized within caveolae [160,161]. In addition, the scaffolding domain of both caveolin-1 and caveolin-3 (but not caveolin-2) binds to PKC, regulating its kinase activity [162,163]. A broad family of proteins, which plays a crucial role in determining how calcium ions are redistributed within cells, is represented by GPCRs [164]. When ligands bind to GPCRs, they initiate a series of intracellular events that ultimately lead to the production of inositol-1,4,5-trisphosphate (IP3). IP3 then binds to its specific receptor within the endoplasmic (or sarcoplasmic) reticulum, causing a rapid release of Ca^2+^ from these compartments. In this context, there is substantial evidence supporting the involvement of caveolae as well as the caveolin-1 CSD in sequestering various GPCR subtypes. Nicotinic acid adenine dinucleotide phosphate (NAADP) is a pivotal signaling molecule, serving as a second messenger [165]. It initiates the release of calcium from lysosomal reservoirs by binding to the calcium channels within the two-pore channel (TPC) family, thereby inducing the efflux of Ca^2+^ into the cytoplasm. Our research has illustrated the participation of caveolae and lipid rafts in mediating NAADP-induced calcium release via the endothelin receptor subtype B (ETB) in smooth muscle cells [113]. Therefore, the role of caveolae in calcium signaling is a fascinating area of research that has provided valuable insights into the intricate mechanisms governing cellular communication. Ongoing research into the precise functions of caveolae, their dynamic interactions with calcium signaling pathways, and their potential as drug targets is likely to yield exciting discoveries.

## 5. Caveolin and Cardiovascular Diseases

### 5.1. Caveolins in Cardiac Function

The cardiovascular system is a complex network responsible for pumping blood throughout the body, ensuring the delivery of essential nutrients and oxygen to various tissues and organs. In recent years, caveolins, primarily caveolin-3, that are abundant in cardiac myocytes, have emerged as a subject of significant interest in the field of cardiology. These proteins have been implicated in various aspects of cardiac function, including the regulation of contractile signaling pathways, ion channel function, and mechanosensory processes within the heart [166]. Overall, the presence of caveolin-3 in cardiac tissue appears essential to provide adequate cardiac protection [167]. It has been reported that increasing caveolin expression in the heart, potentially through PI3K-related pathways, may offer a protective mechanism against injury caused by ischemia and subsequent reperfusion. Mice lacking caveolin-3, exhibiting a total absence of caveolae, display a cardiac dysfunction pattern that closely resembles what is observed in mice lacking caveolin-1 [168,169]. Bryant and collaborators have showed that the lack of caveolin-3 induces cardiac dysfunction in vivo, as well as myocytes hypertrophy, defective t-tubule structure, and decreased t-tubular L-type Ca^2+^ current density in vitro [170]. These findings are consistent with an earlier study by Woodman and collaborators that links the expression of caveolin-3 to the inhibition of the hypertrophic p42/44 MAPK pathway in the heart [168]. Interestingly, although caveolin-3 is the predominant caveolin in cardiomyocytes, caveolin-1 knock-out mice progressively develop a severe cardiomyopathy, show abnormal ventricular wall thickness, hypertrophy, decreased contractility, increased activity of eNOS, and a severely shortened lifespan [171,172,173]. Wright and colleagues conducted a study to explore the role of caveolin-3 in regulating the T-tubular localization of the beta-2 adrenergic receptor (β2AR) and its cAMP signaling, a disrupted pathway in heart failure. The research showed that caveolin-3 selectively influences the spatial compartmentation of β2AR-cAMP responses within the T-tubular compartment. Conversely, overexpressing caveolin-3 in abnormal cardiomyocytes reversed the pathological redistribution of β2AR-cAMP signaling [174]. In a more recent study, Markandeya and colleagues explored the effects of cardiac-specific cavolin-3 loss in adult mouse hearts. While the mice did not display structural remodeling or left ventricular dysfunction, they did exhibit a prolonged QT interval, leading to an elevated risk of ventricular arrhythmia. This suggests that cavolin-3 plays a crucial role in regulating ventricular re-polarization, impacting arrhythmia risk in both mouse and human cardiac cells [175]. Researchers also generated double knockouts mice for caveolin-1 and caveolin-3 genes [176]. These mice are both viable and fertile and exhibit a broader reduction in caveolae in various cell types, including ECs, adipocytes, smooth muscle cells, and skeletal muscle fibers. Interestingly, despite this more extensive depletion, the observed phenotype is like that of mice with single gene knockouts, although more severe. The discovery that the ablation of caveolin protein in the heart is linked to heart dysfunction supports the broader idea of caveolar network playing a crucial role in cardiac protection. Future research on the role of caveolin proteins in heart functions holds significant promise. With coronary artery disease (CAD) and ischemic heart disease continuing to be the leading causes of mortality in industrialized countries, understanding the molecular mechanisms that underlie ischemia-reperfusion injury (I/R injury) is of paramount importance. Exploring how caveolin proteins and other molecular factors contribute to cardiac protection in different contexts, like hypertension, hyperlipidemia, diabetes, aging, and heart failure, can provide valuable insights for the development of targeted therapeutic interventions to mitigate heart-related diseases and complications.

### 5.2. Caveolin-1 and Hypertension

Hypertension, or high blood pressure, is a medical condition characterized by prolonged elevated blood pressure in arteries, posing a significant risk for cardiovascular events. NO, a lipophilic, colorless, and odorless gas with a short half-life in biological fluids, plays a crucial role in various physiological processes within the human body, particularly in the cardiovascular system. One of its key functions is the regulation of blood vessel tone and diameter. Caveolin-1 has been implicated in the pathophysiology of hypertension, mainly through its influence on ECs and vascular smooth muscle cells. Caveolin-1 is known to modulate NO signaling, a key vasodilator, by sequestering eNOS and preventing its activation [177,178]. Elevated caveolin-1 expression in ECs can bind to eNOS and inhibit its activity, resulting in reduced NO production, blood vessel constriction, and increased blood pressure. Conversely, when caveolin-1 expression is low, eNOS operates more efficiently, generating more NO, promoting vasodilation, reducing vascular resistance, and aiding in blood pressure reduction. Consequently, any disruption in this system can contribute to conditions such as hypertension and impaired blood flow, which are known cardiovascular disease risk factors. Wang and colleagues established a hypertension model to elucidate the specific involvement of caveolin-1 in the control of hypertensive vascular remodeling through its influence on the Notch pathway. Their findings suggest that targeting the caveolin-1/Notch1 signaling pathways could offer promising therapeutic avenues for hypertension management [179]. Moreover, a recent study has elucidated the pivotal mechanosensory function of caveolin-1 in the regulation of inflammation induced by pressure in vascular and renal systems [139]. It is worth highlighting how mutations in the proximal portion of the C-terminal domain of caveolin-1 (crucial for the formation of oligomers from monomers), have been observed in patients diagnosed with heritable and idiopathic pulmonary arterial hypertension [27]. In conclusion, caveolin-1 plays a significant role in the pathophysiology of hypertension, shedding light on potential therapeutic targets for managing this complex cardiovascular condition.

## 6. Caveolae and Cellular Senescence

Researchers first noticed the limited lifespan of primary human cells in the 1960s, which led to the development of the concept of senescence [180]. Cellular senescence is a unique biological occurrence observed in somatic cells, marked by a range of distinct features, including irreversible growth arrest, morphological changes, increased senescence-associated β-Galactosidase (SA-β-gal) activity, DNA damage, telomere shortening, and resistance to apoptosis [181,182,183,184]. It also serves as a critical mechanism to ensure proper embryogenesis [185] and has been proposed as a possible therapeutic target in cardiovascular diseases [186]. Recent research has revealed that caveolin-1 regulates multiple pathways involved in senescence, affecting cell signaling, oxidative stress, and the maintenance of cellular structures [187]. Caveolin-1 expression can either promote or inhibit senescence, depending on the cellular context and the signaling pathways involved. Studies have shown that caveolin-1 overexpression has the remarkable ability to inhibit EGF-mediated signaling in senescent fibroblasts. Conversely, the downregulation of caveolin-1 expression through siRNA technology effectively restored EGF-signaling and reactivated cell cycle in senescent fibroblasts [188,189]. Apart from fibroblasts, the influence of caveolin-1 expression on replicative senescence has also been noted in other types of cells, such as human mesenchymal stem cells, macrophages, and chondrocyte [41,190,191,192]. Caveolin-1 has also been associated with stress-induced premature senescence (SIPS). Volonte and colleagues have presented empirical evidence illustrating the direct involvement of caveolin-1 in oxidative stress-induced cellular senescence within human NIH 3T3 cells. This investigation proved that the extent of caveolin-1 expression in NIH 3T3 cells plays a pivotal role in determining whether oxidative stress-mediated senescence or apoptosis is induced [193]. Likewise, it has been reported that smoker patients that were diagnosed coronary atherosclerosis, showed that elevated level of caveolin-1 expression correlated with the expression of the oxidative stress marker 4-hydroxynonenal, resulting in oxidative stress-induced EC senescence, compared with older nonsmokers [194]. From a molecular point of view, among the major molecular pathways by which caveolin-1 mediates SIPS is the p53/p21 signaling pathway [195]. This is due to the ability of caveolin-1 to regulate, directly or indirectly p53 inhibitory molecules, including Mdm2, Ataxia telangiectasia-mutated (ATM), sirtuin 1, nuclear erythroid 2 p45-related factor-2 (Nrf2), and others [187,196,197,198]. Of note, cavin-1, a well-known caveolae associated protein, has been also found increased in senescent fibroblasts, suggesting its role as an additional regulator of cellular senescence acting through the p53/p21 and pathways [199]. Nevertheless, it should be noted that the role of caveolin-1 in cellular senescence is a complex and debated topic. While earlier reported studies suggest that the expression of caveolin-1 can induce cellular senescence, others propose that the downregulation of caveolin-1 is responsible for promoting this process [200]. The morphological and functional alterations seen in ECs when caveolin-1 is silenced by siRNA align with the role of caveolin-1 in diminishing the senescence-like phenotype [201]. This conflicting evidence underscores the intricate interplay between caveolin-1 and cellular senescence, highlighting the need for further research to elucidate the precise mechanisms and conditions under which caveolin-1 exerts its opposite effects on this process.

## 7. Caveolin in Tumor Progression and Stromal Cell Biology

The multifaceted involvement of caveolins in cancer encompasses both pro-tumorigenic and anti-tumorigenic functions, depending on the specific context and type of cancer. On one hand, caveolins can function as tumor suppressors by inhibiting cell proliferation, promoting apoptosis, and regulating the activity of certain signaling pathways. Reasonably, the tumor-suppressive function of caveolin-1 is associated with its ability to engage with numerous proteins situated in the caveolae of the plasma membrane, for example, by sequestering growth-inhibitory proteins, such as p53 and transforming growth factor-β (TGF-β), within caveolae [202]. Nevertheless, a recent study uncovered that the expression of caveolin-1 within the endoplasmic reticulum inhibits the unfolded protein response (UPR) in both in vitro and solid tumors. This repression leads to a reduction in PERK and IRE1α signaling while enhancing cellular vulnerability to endoplasmic reticulum stress and hypoxia. Notably, this tumor-suppressive role relies on the phosphorylation of serine-80 in caveolin-1 [203]. In a study conducted on different cancer cell lines, it has been established that caveolin-1-dependent tumor suppression, especially in the absence of E-cadherin, is linked to reduced HIF1α transcriptional activity via diminished NOS-mediated HIF1α S-nitrosylation [204]. On the other hand, caveolin is known to promote tumor progression by facilitating the activation of pro-oncogenic signaling pathways, such as those mediated by receptor tyrosine kinases [205]. The first evidence highlighting the involvement of caveolin-1 in cell proliferation and tumor cell survival was first observed in Rous sarcoma virus (RSV)-transformed cells [206]. In these cells, the tyrosine phosphorylation of caveolin-1 by protein tyrosine kinase v-SRC resulted in a reduction in both caveolin-1 expression and caveolae [39,207], leading to the transformation process by disrupting normal cell signaling and cellular homeostasis [208]. A similar effect of reduced caveolin-1 has been documented in some tumor types [209,210,211,212]. Conversely, increased caveolin-1 expression has been seen in melanoma cell lines, where caveolin-1 Y14 phosphorylation plays a role in promoting tumor progression [213], or in Ewing’s sarcoma, where its increased expression triggers MMPs-induced metastatic invasion [214,215]. A similar effect of caveolin-1 overexpression on cell proliferation and invasion has been reported in hepatocarcinoma and lung adenocarcinoma [216,217,218,219,220]. This duality of caveolin’s function in cancer highlights its complex nature, making it a subject of ongoing research aimed at better understanding its precise contributions to tumor progression and metastasis.

It is worth mentioning that stromal cells have a crucial role in influencing the development and control of cancer. Found within the tumor microenvironment, these non-cancerous cells, including fibroblasts, mesenchymal stem cells, macrophages, lymphocytes, ECs, and pericytes, are not passive observers but actively engage in the cancer process. Stromal cells can produce growth factors, cytokines, and extracellular matrix components that fuel tumor growth and promote the invasion of cancer cells into surrounding tissues. Within the tumor microenvironment, cancer-associated fibroblasts (CAFs) play a pivotal role in advancing cancer by inhibiting cell apoptosis and promoting the proliferation of cancer cells, induce epithelial-mesenchymal transition, and foster neo angiogenesis [221,222,223,224,225,226]. Likewise, they can also influence immune responses, either suppressing or enhancing the body’s ability to fight cancer [227]. Unlike what is seen in cancer cells, where caveolin-1 exhibits varying effects across different transformed tissues, its expression in the stromal cells in tumor microenvironment seems to follow a more consistent pattern in relation to tumor progression. For instance, the reduced expression of caveolin-1 in CAFs is commonly associated with adverse clinical outcomes, including tumor recurrence, metastasis formation, and resistance to chemotherapy [221,222,228,229,230,231,232]. Sotgia and collaborators have reported that stromal fibroblasts collected from mice lacking caveolin-1 display many resemblances to human CAFs [233]. Overall, multiple research studies concur with the hypothesis that the reduction in caveolin-1 in the stromal microenvironment of breast cancer indicates an unfavorable clinical prognosis [232,234,235]. A diminished level of caveolin-1 expression has also been shown in stromal cells across several types of cancers, such as prostate, gastric, and melanoma [228,230,236,237]. The expression of caveolin proteins in tumor stromal cells exerts profound effects on cancer development and therapeutic responses, making them attractive targets for future research and potentially novel anti-cancer therapies.

## 8. Caveolin and Neurodegenerative Disorders

Neurodegenerative disorders (NDDs) represent a class of debilitating diseases characterized by the progressive degeneration of neurons in the nervous system. These disorders, which include Alzheimer’s disease (AD), Parkinson’s disease (PD), and amyotrophic lateral sclerosis (ALS), have far-reaching implications for the affected individuals and their families, making it imperative to decipher the intricate mechanisms underlying their pathogenesis. Emerging research has suggested that caveolins play a pivotal role in the pathogenesis of NDDs by influencing various cellular processes, including protein aggregation, oxidative stress, and synaptic dysfunction. Studies conducted on rats have shown that caveolin-1 is extensively distributed across various brain compartments [238], and all three caveolin proteins are expressed in brain ECs and astrocytes. This observation marks the first instance of caveolin-3 expression occurring beyond the confines of the muscular system [21]. Wu et al. showed that caveolae-like domains are prominently present in neuronal plasma membranes and have a high concentration of signaling molecules, including tyrosine kinases [239]. Consistent with these findings, a growing body of evidence has been gathered to support the regulatory role of caveolins in intracellular signaling within neurons [240]. Biochemical analysis has uncovered the presence of both caveolin-1 and caveolin-2 in dorsal root ganglion and pheochromocytoma PC12 cells, with their levels increasing following treatment with nerve growth factor (NGF) [241]. In more recent studies, Wang and colleagues found that an increased expression of caveolin-1 in human neuronal progenitor cells promotes the growth of axons. Conversely, when caveolin-1 phosphorylation is inhibited, it negatively affects both the differentiation of neuronal progenitor cells and the growth of axons. An effect due to the blunting of caveolin-1 mediated Rac1/Cdc42 signaling, a crucial signal regulating filopodia and neurite outgrowth [123]. In a different study, Koh et al. showed that caveolin-1 is found within lipid rafts situated at nerve terminals in primary cultured hippocampal neurons and plays a vital role in presynaptic processes. The authors proved that caveolin-1 silencing using siRNA significantly impairs synaptic transmission as well as decrease the rate of both synaptic vesicle exocytosis and endocytosis [242]. Caveolin-1 plays a crucial neuroprotective role in the brain, as shown by studies conducted on caveolin-1 knockout mice. A study by Jasmin and collaborators [243] revealed that the levels of caveolin-1 and 2 increased during cerebral ischemia. Conversely, when caveolin-1 was genetically removed, it was associated with larger areas of brain tissue damage, impaired angiogenesis, and increased cell death in comparison to wild-type mice. These findings support the idea that caveolin-1 serves a neuroprotective function. Other studies have further highlighted the importance of this protein in maintaining and regulating the permeability of the blood–brain barrier [244].

Cholesterol metabolism in the brain is crucial for supporting normal brain function, and imbalances in this process can have significant implications for neurological health, including the development of neurodegenerative diseases like AD. Due to their role in maintaining cellular cholesterol balance and overall lipid homeostasis, caveolins and caveolae are expected to play a significant role in cholesterol metabolism in the nervous system. Scientific evidence suggest that the interaction between cholesterol levels and the processing of the amyloid precursor protein (APP) plays a significant role in AD [245]. It is widely recognized that only about 25% of the cholesterol in the bloodstream can penetrate the blood–brain barrier [246]. As a result, the brain is self-sustaining and capable of producing the cholesterol it needs, thanks to a finely controlled cholesterol metabolism and supporting glial cells, primarily astrocytes. One of the most relevant mechanisms that is believed to contribute to the development of AD is the accumulation of the beta-amyloid protein in the brain, resulting in the formation of plaques that disrupt communication between nerve cells and trigger inflammation. Notably, beta-amyloid protein has been found localized in a detergent-insoluble glycolipid-enriched membrane domain where it is supposed to be processed from the cleavage of the larger amyloid precursor protein (APP) [247,248]. Moreover, caveolin-1 is recognized for its physical interaction with APP through its C-terminal tail, and when caveolin-1 is overexpressed, it reduces the proteolysis of APP that is mediated by γ-secretase [247,249]. Beyond the role of caveolin-1, some studies have shown that caveolin-2 may play roles in neurodegenerative diseases, particularly AD. It is worth noting that the expression of caveolin-2 decreases when activated by the Rac1b signaling pathway, resulting in the formation of neurofibrillary tangles (NFTs) in neurons [250]. NFTs are considered a neuropathological hallmark of AD characterized by a dysfunctional cytoskeleton, along with the loss of microtubules and tubulin-associated proteins [251]. Like caveolin-1, caveolin-3 has been discovered to be physically linked with APP, and it showed a notable increase in reactive astrocytes around senile plaques in tissue derived from patients with AD [252]. These findings supply support for the involvement of caveolin-1 and -3 in the regulation of amyloid production and open new avenues to decipher the amyloid biogenesis in neurodegenerative disorders.

There is a growing body of evidence that shows a strong link between increased levels of caveolin-1 and the key molecular aspects of PD. Caveolin-1 plays a pleiotropic role in PD, showing both positive and negative impacts on its development. The analysis of human neuroblastoma cells has shown that overexpressing α-Synuclein, a presynaptic neuronal protein, disrupts the ERK signaling through an elevation in caveolin-1 levels resulting in reduced neurite outgrowth and cell adhesion ultimately contributing to the development of PD [253]. Moreover, the increase in α-Synuclein inhibits cell survival by upregulating the caveolin-1 level [254]. However, neither of these studies was able to pinpoint the exact molecular mechanisms through which α-Synuclein regulates the expression of caveolin-1. A similar mechanism leading to caveolin-1 upregulation is seen in the reduced expression of parkin, belonging to a group of genes whose mutations have been clearly linked to PD [255]. This leads to the inhibition of caveolin-1 degradation via the proteasome-dependent pathway, resulting in an imbalanced cholesterol level, the alteration of membrane fluidity, and dysfunctional lipid raft-dependent endocytosis, which ultimately contribute to PD [256]. Conversely, research by Cataldi et al. demonstrated that 1-Methyl-4-phenyl-1,2,3,6-tetrahydropyridine (MPTP), a common lipophilic substance used to induce PD reduces caveolin-1 expression, thus contributing to PD development [257]. In addition, a deficiency in DJ-1 (also known as Protein deglycase or PARK 7), a gene associated with familial PD, induces the downregulation of caveolin-1 expression through its ability to regulate flotillin-1, leading to dysfunctional cholesterol metabolism, lipid raft-mediated endocytosis, and the impairment of glutamate uptake into astrocytes [258]. Of note, a comparable situation has also been seen by the same authors in DJ-1 knockout mouse embryonic fibroblasts, showing that the mechanism through which DJ-1 regulates the expression of caveolin-1 and flotillin-1 is not exclusively limited to astrocytes. Furthermore, a diminished presence of caveolin proteins has been documented as diminishing the protective benefits of natural tocotrienols, a member of the vitamin E family, and a natural compound, which has shown its potential in averting neurodegeneration and lowering the risk of PD [259,260]. The growing body of evidence linking caveolin proteins to the fundamental aspects of nervous system function makes them promising candidates for future research in the pathophysiology of neurodegenerative disorders. This research has the potential to substantially illuminate the underlying mechanisms and, ultimately, contribute to the development of innovative therapeutic approaches.

## 9. Conclusions

Targeting caveolins and caveolae holds great promise in the field of cell signaling and disease management. Their pivotal roles in various diseases, from cardiovascular conditions to cancer and neurodegenerative disorders, have opened new avenues for innovative therapeutic approaches. Moreover, targeting caveolae, caveolin proteins, and associated signaling molecules such as src kinase, cavins, lipid metabolism enzymes, eNOS, and ion channels holds promise for addressing various human diseases. On the other hand, the complex and multifaceted nature of the caveolar network poses challenges, from their intricate biogenesis to their context-dependent effects on disease progression. Therefore, future research should focus on uncovering the precise mechanisms behind these structures and harnessing their potential as therapeutic targets, thereby advancing medical science.

## Figures and Tables

**Figure 1 cells-12-02680-f001:**
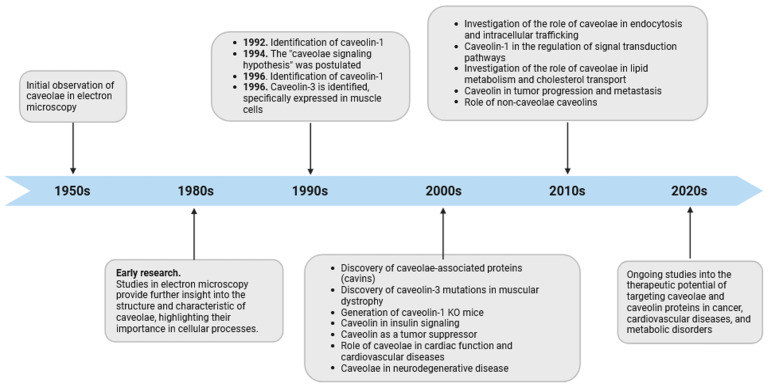
Timeline of caveolae research. The diagram depicts significant observations in caveolae biology from their discovery to the present, emphasizing their suggested participation in diverse cellular processes. It is important to highlight that the “caveolae signaling hypothesis” and the participation of caveolin domains in protein–protein interactions should be re-evaluated in light of recent cryoEM studies on caveolin, as discussed in this review article.

**Figure 2 cells-12-02680-f002:**
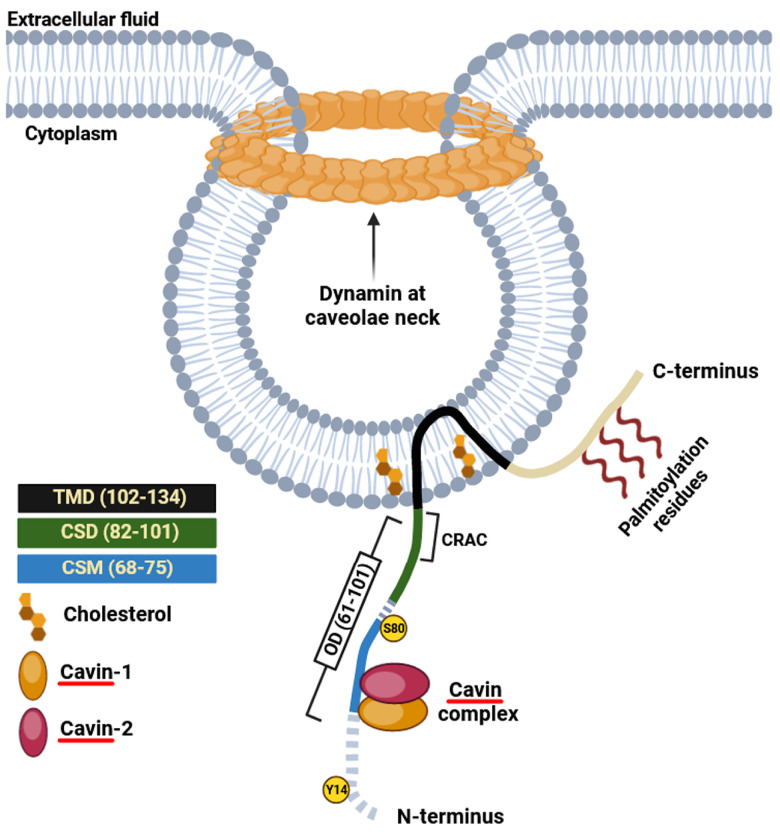
Schematic illustration of caveolin-1. CRAC sequence “VTKYWFYR” spanning residues 94–101; CSD, caveolin scaffolding domain; CSM, caveolin signature motif “FEDVIAEP”; OD, oligomerization domain; TDM, transmembrane domain. Two phosphorylation sites at tyrosine 14 (Y14) and serine 80 (S80) are indicated. Cavin proteins directly interact with caveolins, forming a stable complex that is critical for the formation of caveolae and their localization in the plasma membrane.

**Figure 3 cells-12-02680-f003:**
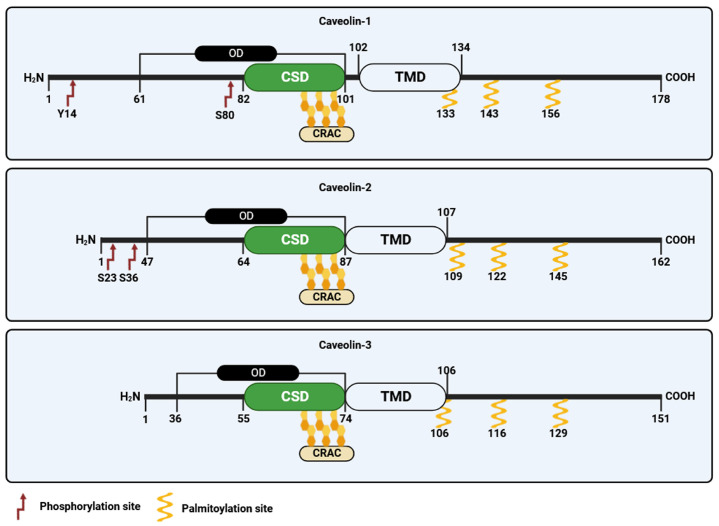
Schematic representation of the caveolin family of proteins. CRAC, cholesterol recognition/interaction amino acid consensus; OD, oligomerization domain; TDM, transmembrane domain; CSD, caveolin scaffolding domain.

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
