# Peer review of "Unraveling the Cave: A Seventy-Year Journey into the Caveolar Network, Cellular Signaling, and Human Disease"

_cells, 2023, doi:10.3390/cells12232680_

Round 1
Reviewer 1 Report
Comments and Suggestions for Authors
This is a well-written comprehensive review of the historical advancements in caveolae research, covering the role of caveolar network in different tissues and in human diseases.
I think that the review will gain a more rounded up view if the author would show the role caveolae in cardiovascular biology. For example, the review will benefit if changes in ventricular cardiomyocytes in cardio-specific Cav3KO mice would be discussed. In addition, the role of caveolae in beta adrenergic receptor compartmentation may be discussed.
Author Response
Reviewer 1
- I think that the review will gain a more rounded up view if the author would show the role caveolae in cardiovascular biology. For example, the review will benefit if changes in ventricular cardiomyocytes in cardio-specific Cav3KO mice would be discussed. In addition, the role of caveolae in beta adrenergic receptor compartmentation may be discussed.
Author’s response. I express gratitude to the reviewer for the valuable suggestion. Consequently, I have included an additional paragraph (highlighted in blue) aimed at providing a more in-depth analysis of the impact of caveolin-3 loss in cardiomyocytes and the function of caveolae in the compartmentalization of the beta-adrenergic receptor. Please refer to the recently added paragraph spanning from line 525 to 536 (Section 4.5 Caveolins in Cardiac Function), inclusive of two newly included references (174 and 174).

Reviewer 2 Report
Comments and Suggestions for Authors
Caveolae are intriguing structures whose function has been studied for many years. The purpose of this review is to provide an overview of the history of the field and summarize the physiological importance of caveolae and their roles in disease. I appreciate the effort the author put into assembling this information. The manuscript is generally well written and organized. However, I have several major concerns about the content.
1. The review glosses over several major controversies in the field. Specific examples include the role of caveolae in endocytosis 1 2, whether/how caveolae serve as scaffolds for signaling molecules 3 4, 5, and whether caveolae are composed primarily of their structural and accessory proteins or can also contain other proteins 2 6. These types of studies lay the foundation for understanding how caveolae function at a mechanistic level. It is thus essential that these points are addressed appropriately.
2. The review briefly acknowledges the recently determined structure of caveolin-1 7, yet the figures depict the structure of caveolin in the same way as it has been shown for many years. This is misleading and should be corrected to reflect the current state of knowledge. One would also think that the elucidation of the structure would qualify as a milestone in the field.
3. The review contains a significant number of unreferenced statements. Especially for a review, the reader expects all of the information provided to include citations. There are far too many to list, but here are a few examples:
a. Line 100: “Notably a cholesterol recognition/interaction amino acid consensus (CRAC) “V_T_K_Y_W_F_Y_R_” _spans residues 94-101 within 101 the CSD, allows caveolin-1 to interact with cholesterol molecules.”
b. Line 231: “A widely accepted hypothesis suggests that caveolae generally do not participate in endocytosis.”
c. Line 629: “Scientific and epidemiological evidence suggest that the interaction between 629 cholesterol levels and the processing of the amyloid precursor protein (APP) plays a significant role in AD.”
Other concerns
4. There are many excellent reviews on caveolae and caveolins. I recommend that the author point to more of them to provide a more balanced view of the existing literature.
References
1. Matthaeus, C.; Taraska, J. W., Energy and dynamics of caveolae trafficking. Front Cell Dev Biol 2020, 8, 614472.
2. Shvets, E.; Bitsikas, V.; Howard, G.; Hansen, C. G.; Nichols, B. J., Dynamic caveolae exclude bulk membrane proteins and are required for sorting of excess glycosphingolipids. Nat Commun 2015, 6, 6867.
3. Collins, B. M.; Davis, M. J.; Hancock, J. F.; Parton, R. G., Structure-based reassessment of the caveolin signaling model: do caveolae regulate signaling through caveolin-protein interactions? Dev Cell 2012, 23 (1), 11-20.
4. Byrne, D. P.; Dart, C.; Rigden, D. J., Evaluating caveolin interactions: do proteins interact with the caveolin scaffolding domain through a widespread aromatic residue-rich motif? PLoS ONE 2012, 7 (9), e44879.
5. Jung, W.; Sierecki, E.; Bastiani, M.; O'Carroll, A.; Alexandrov, K.; Rae, J.; Johnston, W.; Hunter, D. J. B.; Ferguson, C.; Gambin, Y.; Ariotti, N.; Parton, R. G., Cell-free formation and interactome analysis of caveolae. J Cell Biol 2018, 217 (6), 2141-2165.
6. Ludwig, A.; Howard, G.; Mendoza-Topaz, C.; Deerinck, T.; Mackey, M.; Sandin, S.; Ellisman, M. H.; Nichols, B. J., Molecular composition and ultrastructure of the caveolar coat complex. PLoS Biol 2013, 11 (8), e1001640.
7. Porta, J. C.; Han, B.; Gulsevin, A.; Chung, J. M.; Peskova, Y.; Connolly, S.; Mchaourab, H. S.; Meiler, J.; Karakas, E.; Kenworthy, A. K.; Ohi, M. D., Molecular architecture of the human caveolin-1 complex. Sci Adv 2022, 8(19), eabn7232.
Comments on the Quality of English LanguageIn general, the manuscript is well written and organized. However, there are numerous grammatical and typographical errors. Furthermore, the paragraphs are very long. Splitting the text up into smaller paragraphs would make the text easier to read.
Author Response
Reviewer 2
- The review glosses over several major controversies in the field. Specific examples include the role of caveolae in endocytosis 12, whether/how caveolae serve as scaffolds for signaling molecules 3 4, 5, and whether caveolae are composed primarily of their structural and accessory proteins or can also contain other proteins 2 6. These types of studies lay the foundation for understanding how caveolae function at a mechanistic level. It is thus essential that these points are addressed appropriately.
Author’s response. I wish to express my gratitude to the reviewer for providing invaluable suggestions. First, I concur with her/his observation that, as articulated in the prior version of the manuscript, the elucidation of caveolae's involvement in the endocytosis mechanism was unclear and somewhat confusing. To address this concern, I have extensively revised Section 4.1 "Caveolae and Caveolin in Endocytosis and Transcytosis." I hope that these modifications contribute to a clearer comprehension of this subject matter. New references, as suggested by the reviewer, have been incorporated into the respective section. Please note that, all modifications are denoted by blue text, while deleted portions are identified by red strikethrough text.
- The review briefly acknowledges the recently determined structure of caveolin-1, yet the figures depict the structure of caveolin in the same way as it has been shown for many years. This is misleading and should be corrected to reflect the current state of knowledge. One would also think that the elucidation of the structure would qualify as a milestone in the field.
Author’s response. I apologize to the reviewer for the confusion. The purpose was to present the current arrangement of caveolin-1 domains and emphasize recent research exploring the three-dimensional organization of the protein and its interaction with the plasma membrane. In this context, Section 3.1, titled "Caveolin-1," has been restructured, as outlined in the revised manuscript from line 108 onward.
- The review contains a significant number of unreferenced statements. Especially for a review, the reader expects all of the information provided to include citations. There are far too many to list, but here are a few examples:
- Line 100: “Notably a cholesterol recognition/interaction amino acid consensus (CRAC) “V_T_K_Y_W_F_Y_R_” spans residues 94-101 within 101 the CSD, allows caveolin-1 to interact with cholesterol molecules.”
Author’s response. A new reference has been included at line 108.
- Line 231: “A widely accepted hypothesis suggests that caveolae generally do not participate in endocytosis.”.
Author’s response. I concur with the reviewer's assertion that this sentence is improper, causing confusion. Therefore, it was removed from the original manuscript. In addition, please note that the Section 4.1 “Caveolae and caveolin in endocytosis and transcytosis”, has been extensively rearranged.
- Line 629: “Scientific and epidemiological evidence suggest that the interaction between cholesterol levels and the processing of the amyloid precursor protein (APP) plays a significant role in AD.”
Author’s response. I apologize to the reviewer for not correctly inserting the citation relating to this statement. In this regard, the corrected version of the manuscript contains the appropriate reference (see new ref 245).
Other concerns
- There are many excellent reviews on caveolae and caveolins. I recommend that the author point to more of them to provide a more balanced view of the existing literature.
Author’s response. I am aware that there are many excellent quality reviews in the literature. In this regard, I have done my best to mention the most recent and most significant ones. However, if the reviewer prefers to suggest others, I will be happy to include them in the citation list.
- Comments on the Quality of English Language. In general, the manuscript is well written and organized. However, there are numerous grammatical and typographical errors. Furthermore, the paragraphs are very long. Splitting the text up into smaller paragraphs would make the text easier to read.
Author’s response. I apologize to the reviewer for the grammatical errors in the text. In this regard, I thank the reviewer for pointing them out. Grammatical and typographical errors have been identified and corrected.
Regarding the length of the paragraphs, I beg to differ with the reviewer on this point. I would like to point out that the discussed topics have also been covered in sub-paragraphs to make reading easier. It would be difficult to reduce their length without also reducing the information contained in each of them. Perhaps the only longer paragraph is the one titled "Caveolin and neurodegenerative disorders." But I find it difficult to divide it into sub paragraphs.

Reviewer 3 Report
Comments and Suggestions for Authors
This is an excellent and comprehensive review of caveolin-1/caveolae that shall be of utmost interest to the overall scientific community. It succeeds in providing up to date integration of the molecular, structural, signaling and pathophysiological roles of cav-1/CAV. It is acceptable for publication provided the author considers the following recommendations that can expand the comprehensiveness and state of knowledge of the field and make some minor corrections.
Although not a primary focus of the review, the author may consider providing additional summary sentences on the roles of caveolins localized outside of caveolae. In this respect, special consideration can be given to the review by Pol A, Morales-Paytuví F, Bosch M, Parton RG. Non-caveolar caveolins - duties outside the caves. J Cell Sci. 2020;133(9):jcs241562. This reference also provides more complete information than the quoted refences 22 and 23 of the manuscript. Accordingly, the emergence of non-caveolar roles of caveolins can be incorporated into Figure 1. Timeline of major discoveries in the field of caveolae research.
The manuscript gives an excellent overview of caveolae-caveolins functional roles and structural aspects, including mentioning of the cavin family of proteins. Due to the documented role of cavins in caveolae biogenesis and stability Section 3.3. The cavin family of proteins (line 192) can include reference to summary findings of the relationship between-caveolins-cavins-caveolae and pathophysiological roles. In this context the author quotes Kovtun, et al. (2015). Cavin family proteins and the assembly of caveolae. J Cell Sci 2015, 128, 1269-1278; and, may consider adding the reference of Liu L. Lessons from cavin-1 deficiency. Biochem Soc Trans. 2020;48(1):147-154 to include a summary of postulated cavin-1 pathophysiological roles. Author may also consider at this juncture to insert succinctly the information provided by the author on “dolines” in lines 362-364 (REF 142).“ Given these considerations are accepted, Figure 2 can be edited to include a small representation of cavins in the caveolae model. Irrespectively, the legend of Figure 2 must be revised to include: “FEDVIAEP” caveolin signature motif (CSM, residues 68 to 75).
In the Conclusions section the sentence in lines 690-693 strikes as too long and “overwhelming”: “Additionally, the development of drugs targeting caveolae, caveolin proteins, and the myriad of signaling molecules they regulate, including src kinase, cavins, lipid metabolism enzymes, eNOS, and ion channels, just to mention a few, may offer potential solutions for numerous human diseases“ Author may simply edit the sentence. Alternatively, Figure 2 can be modified to portray cav-1-positive CAV and associated structural and signaling molecules as a “CAV complex or network or CAV structural/functional scaffold”, clearly distinct from the once proposed and withdrawn concept of the caveosome from the cellular pain of view. In lights of the complexity of the topic defining refering in the body of the review and conclusions to an unitary model of the status of caveolin-1 positive caveolae and associated structural and signaling molecules shall be of significant benefit to the field.
Comments on the Quality of English Language
Minor corrections are:
· typo line 140: Like caveolin-1, caveolin-2 shows
· consider editing line 214 as follows: In turn, transcytosis refers to the transport of materials from one side…
· typo line 239: …caveolin-1-deficient mice…
· typo line 543: …different cancer cell lines
· typo line 563: …that stromal cells …
Author Response
Reviewer 3
- Although not a primary focus of the review, the author may consider providing additional summary sentences on the roles of caveolins localized outside of caveolae. In this respect, special consideration can be given to the review by Pol A, Morales-Paytuví F, Bosch M, Parton RG. Non-caveolar caveolins - duties outside the caves. J Cell Sci. 2020;133(9):jcs241562. This reference also provides more complete information than the quoted refences 22 and 23 of the manuscript. Accordingly, the emergence of non-caveolar roles of caveolins can be incorporated into Figure 1. Timeline of major discoveries in the field of caveolae research.
Author’s response. I express my gratitude to the reviewer for the valuable suggestions. I concur that the primary emphasis of this work does not center on the role of non-caveolar caveolin. Nevertheless, I believe it is beneficial to provide a concise commentary on this subject. Consequently, I have incorporated a brief paragraph spanning lines 69 to 77, accompanied by two additional references (24 and 25). A new entry “Non-caveolar caveolins” has been added to Figure 1 in the 2010s box. I would like to point out that the entire figure 1 has also been improved in style compared to the previous version of the manuscript.
- The manuscript gives an excellent overview of caveolae-caveolins functional roles and structural aspects, including mentioning of the cavin family of proteins. Due to the documented role of cavins in caveolae biogenesis and stability Section 3.3. The cavin family of proteins (line 192) can include reference to summary findings of the relationship between-caveolins-cavins-caveolae and pathophysiological roles. In this context the author quotes Kovtun, et al. (2015). Cavin family proteins and the assembly of caveolae. J Cell Sci 2015, 128, 1269-1278; and, may consider adding the reference of Liu L. Lessons from cavin-1 deficiency. Biochem Soc Trans. 2020;48(1):147-154 to include a summary of postulated cavin-1 pathophysiological roles.
Author’s response. I thank the reviewer for the provided suggestion. In response to this recommendation, a concise paragraph has been incorporated, starting from line 222. In accordance with the request, I have succinctly expounded upon the pathophysiological role of cavin-1 in this section, including a new reference n 68.
- Author may also consider at this juncture to insert succinctly the information provided by the author on “dolines” in lines 362-364 (REF 142).“ Given these considerations are accepted, Figure 2 can be edited to include a small representation of cavins in the caveolae model. Irrespectively, the legend of Figure 2 must be revised to include: “FEDVIAEP” caveolin signature motif (CSM, residues 68 to 75).
Author’s response. I thank the reviewer for the insightful suggestion provided. In response, I have incorporated a more comprehensive description elucidating the structure and significance of "dolines" within this context (line 451-455). Additionally, in accordance with the reviewer's recommendation, the legend for Figure 2 has been revised.
- In the Conclusions section the sentence in lines 690-693 strikes as too long and “overwhelming”: “Additionally, the development of drugs targeting caveolae, caveolin proteins, and the myriad of signaling molecules they regulate, including src kinase, cavins, lipid metabolism enzymes, eNOS, and ion channels, just to mention a few, may offer potential solutions for numerous human diseases“ . Author may simply edit the sentence.
Author’s response. I apologize to the reviewer for this sentence. As suggested, I have rephrased this passage, hopefully, into a more readable form, as indicated by the new text highlighted in blue in the “conclusions” section.
- Alternatively, Figure 2 can be modified to portray cav-1-positive CAV and associated structural and signaling molecules as a “CAV complex or network or CAV structural/functional scaffold”, clearly distinct from the once proposed and withdrawn concept of the caveosome from the cellular pain of view. In lights of the complexity of the topic defining refering in the body of the review and conclusions to an unitary model of the status of caveolin-1 positive caveolae and associated structural and signaling molecules shall be of significant benefit to the field.
Author’s response. If my comprehension is accurate, the reviewer proposes a modification to Figure 2 by incorporating additional details pertaining to the signal molecules that are regulated by the caveola. I would like to point out that the goal here was to create a concise and straightforward representation of the key players in the caveolar biogenesis. I believe that introducing additional graphical information could pose challenges to clarity.
- Comments on the Quality of English Language
Minor corrections are:
- typo line 140: Like caveolin-1, caveolin-2 shows
- consider editing line 214 as follows: In turn, transcytosis refers to the transport of materials from one side…
- typo line 239: …caveolin-1-deficient mice…
- typo line 543: …different cancer cell lines
- typo line 563: …that stromal cells …
Author’s response. I thank the reviewer for pointing out these small typos. They have been reviewed and corrected within the text.

Round 2
Reviewer 2 Report
Comments and Suggestions for Authors
The author has addressed the majority of my previous concerns by adding several additional sections to the review. The new text more accurately reflects ongoing controversies in the field and provides an important set of caveats for the reader to consider. Overall, there is a lot of useful information summarized here that will be useful for the field.
My one remaining concern is in regard to Figure 1. While I understand that the author wishes to keep this as a schematized drawing, the concern is that many aspects of the model of caveolin shown here have recently been shown to be inaccurate based on the cryoEM structure and the figure thus remains misleading in its current form. The author argued that the intent was depict the domain structure of caveolin. However, this is shown in Figure 2 and it seems unnecessary to repeat this information twice. If the author is set on keeping the figure as is, it is essential that they add a sentence to the figure legend clearly stating that this model is highly oversimplified and is likely incorrect based on recent structural evidence.
Author Response
- My one remaining concern is in regard to Figure 1. While I understand that the author wishes to keep this as a schematized drawing, the concern is that many aspects of the model of caveolin shown here have recently been shown to be inaccurate based on the cryoEM structure and the figure thus remains misleading in its current form. The author argued that the intent was depict the domain structure of caveolin. However, this is shown in Figure 2 and it seems unnecessary to repeat this information twice. If the author is set on keeping the figure as is, it is essential that they add a sentence to the figure legend clearly stating that this model is highly oversimplified and is likely incorrect based on recent structural evidence.
Author’s response. I want to express my gratitude to the reviewer for diligently analyzing the text. Any modification contributing to an enhancement in the quality of the work, particularly the inclusion of accurate information on the topics covered, is crucial and appreciated. I fully understand this additional observation made by the reviewer. The scheme shown in figure 1 has the aim of reporting in a simplified way the most significant "discoveries" obtained over the years in the field of caveolae. I personally thought this could be useful especially for readers unfamiliar with this field. Considering that I personally have no issue with removing the figure from the work, I would like to suggest the following changes and, consequently, keep it in the manuscript. Alternatively, I am open to excluding the figure from the manuscript. The changes proposed are the following:
- in the box relating to the 2010s, the words "Discovery" have been replaced with "Investigation."
The figure legend has been modified to emphasize that the diagram is extremely simplified and must be read in light of the recent discoveries discussed in this manuscript. Changes are highlighted in blue in the legend.
